# Function of Graphene Oxide as the “Nanoquencher” for Hg^2+^ Detection Using an Exonuclease I-Assisted Biosensor

**DOI:** 10.3390/ijms23116326

**Published:** 2022-06-05

**Authors:** Ting Sun, Xian Li, Xiaochuan Jin, Ziyi Wu, Xiachao Chen, Jieqiong Qiu

**Affiliations:** 1College of Life Sciences and Medicine, Zhejiang Sci-Tech University, Hangzhou 310018, China; z1178390272@163.com (T.S.); adadallen@163.com (X.L.); jxc20012022@163.com (X.J.); ziyiwu1999@163.com (Z.W.); 2School of Material Sciences & Engineering, Zhejiang Sci-Tech University, Hangzhou 310018, China; chenxiachao@zstu.edu.cn

**Keywords:** positively charged graphene oxide (pGO), exonuclease I, fluorescence quencher, hairpin structure, T–Hg^2+^–T

## Abstract

Graphene oxide is well known for its excellent fluorescence quenching ability. In this study, positively charged graphene oxide (pGO25000) was developed as a fluorescence quencher that is water-soluble and synthesized by grafting polyetherimide onto graphene oxide nanosheets by a carbodiimide reaction. Compared to graphene oxide, the fluorescence quenching ability of pGO25000 is significantly improved by the increase in the affinity between pGO25000 and the DNA strand, which is introduced by the additional electrostatic interaction. The FAM-labeled single-stranded DNA probe can be almost completely quenched at concentrations of pGO25000 as low as 0.1 μg/mL. A simple and novel FAM-labeled single-stranded DNA sensor was designed for Hg^2+^ detection to take advantage of exonuclease I-triggered single-stranded DNA hydrolysis, and pGO25000 acted as a fluorescence quencher. The FAM-labeled single-stranded DNA probe is present as a hairpin structure by the formation of T–Hg^2+^–T when Hg^2+^ is present, and no fluorescence is observed. It is digested by exonuclease I without Hg^2+^, and fluorescence is recovered. The fluorescence intensity of the proposed biosensor was positively correlated with the Hg^2+^ concentration in the range of 0–250 nM (R^2^ = 0.9955), with a seasonable limit of detection (3σ) cal. 3.93 nM. It was successfully applied to real samples of pond water for Hg^2+^ detection, obtaining a recovery rate from 99.6% to 101.1%.

## 1. Introduction

Water-soluble mercury(II) ion (Hg^2+^), as one of the most familiar environmental pollutants, is a toxic heavy metal that can exist in metallic, inorganic, and organic forms, especially in freshwater and marine ecosystems [1]. After prolonged exposure, it is extremely toxic to the brain, kidney, and other organs of organisms at very low mercury(II) concentrations [2]. The accumulation of heavy metals can occur in animal and human bodies via the food chain and damage the reproductive, gastrointestinal, and cardiovascular systems. Based on the guidelines of the United States Environmental Protection Agency (EPA), the Ministry of Health of the People’s Republic of China (MOH), and the World Health Organization (WTO), the maximum mercury(II) concentration in drinking water should be as low as 10 nM [3].

Currently, many traditional techniques have been developed for Hg^2+^ analysis and detection, including inductively coupled plasma–optical emission spectrometry (ICP-OES) [4], chemical vapor generation–inductively coupled plasma–optical emission spectrometry (CVG-ICP-OES) [5], inductively coupled plasma–mass spectrometry (ICP-MS) [6], cold vapor atomic absorption spectroscopy (CVAAS) [7], atomic fluorescent spectroscopy [8], and electrochemical methods. Hg^2+^ can be detected at pM concentrations by most of the abovementioned methods. However, high-cost, complex sample preparation and professional operation are also needed. Therefore, it is essential to explore rapid, specific, sensitive, cost-efficient, convenient, and real-time biosensors instead of traditional approaches for monitoring heavy metal ions.

Fluorescence-based methods have been widely used as potential techniques for Hg^2+^ detection. In view of the sensitivity improvement for Hg^2+^ detection, an increasing number of researchers have paid attention to signal amplification in DNA-based strategies by exonucleases, including exonuclease I and III (Exo I and III) [9,10,11,12,13]. For example, Exo I is a 3′–5′ exonuclease that can cleave single-stranded DNA (ssDNA) without sequence-dependence [14,15]. Hg^2+^ can promote the formation of a DNA duplex (dsDNA) via T–Hg^2+^–T formation, which is not allowed to be digested by Exo I. However, Exo I can hydrolyze ssDNA. Thus, there is potential for applying Exo I in Hg^2+^ detection on the basis of DNA-based signal amplification strategies.

However, traditional fluorescent DNA probes, such as Taqman probes, molecular beacons (MBs), and scorpions, cannot meet the requirements [16,17] because no free 3′-OH is present. For this reason, a label-free assay has been developed based on the fluorescence “turn-on” caused by dye intercalation into special DNA structures [18,19,20]. However, such label-free dyes have a non-negligible fluorescent background.

To solve these problems, guanine bases [16] and nanostructures [21] have been used as quenchers for DNA probes. Various nanostructures, such as gold nanoparticles (AuNPs) [22,23], single-walled carbon nanotubes (SWCNTs) [24], graphene oxide (GO) [25], fullerene (C_60_) [26,27], multiwalled carbon nanotubes (MWCNTs) [9], and positive carbon dots (P-CDs) [28], have been successfully used as nanoquenchers for mercury(II) ion detection. As a two-dimensional (2D) material, GO exhibits high-efficiency fluorescence quenching, good water dispersibility, low cost, and various surface modifications. Therefore, it is frequently used in biosensors [29]. Previous studies have confirmed that ssDNA is labelled with a fluorescent dye, which can be quenched by GO due to fluorescence resonance energy transfer (FRET) [12,30]. This result is attributed to the hydrogen bond and π–π stacking caused by nucleobases and GO, which make FRET more efficient. The fluorescence quenching efficiency of GO is dependent on the GO quantity used. A high concentration of GO can limit its application in Hg^2+^ detection, e.g., in cells. Therefore, it is necessary to improve the fluorescence quenching ability and efficiency of GO. In addition, the fluorescence quenching efficiency of GO can be increased by partially reducing graphene oxide due to the increase in π–π stacking interaction [25].

In this work, positively charged graphene oxide (pGO25000) was synthesized by grafting polyetherimide (PEI) onto GO nanosheets by a carbodiimide reaction. The first use of positively charged pGO25000 as an efficient fluorescence quencher was demonstrated. Compared to GO, the fluorescence quenching efficiency of pGO25000 can be enhanced by the positively charged surface that allows attraction of the negatively charged DNA strands via electrostatic interaction. Based on the special property of pGO25000, a FAM–ssDNA probe was designed for the highly selective and ultrasensitive detection of mercury(II) ions with the assistance of the Exo I enzyme under mild conditions.

## 2. Results and Discussion

### 2.1. Strategy for Ultrasensitive Detection of Hg^2+^

A FAM–ssDNA probe was designed for the highly sensitive and highly selective detection of Hg^2+^, with pGO25000 being a fluorescence nanoquencher and Exo I being a special enzyme for hydrolyzing ssDNA in the 3′→5′ direction, as shown in Figure 1. GO, as a fluorescence quencher, can quench fluorescent dye via FRET when GO and fluorescent dye are sufficiently close to each other. Considering the binding affinity of ssDNA to GO, which results from π–π stacking and hydrogen bonding between ssDNA and GO, the dye-labelled ssDNA probes were designed for Hg^2+^ detection based on the GO fluorescence quenching ability. Positively charged GO (PEI-GO) has been reported as a fluorescence quencher of anionic dyes (i.e., Merocyanine 540) via electrostatic interactions [31]. However, dye-labelled DNA probes have never been reported as fluorescence quenchers. In this work, pGO25000 was synthesized by grafting PEI (M.W. = 25,000) onto GO nanosheets, which can selectively bind to ssDNA/dsDNA at very low concentrations. Thus, the fluorescence quenching efficiency was enhanced based on the additional electrostatic attraction between the phosphate group and positively charged PEI. The fluorescence was almost quenched for the FAM–ssDNA probe by pGO25000; however, FAM–dsDNA can be also quenched by pGO25000. To solve this problem, enzyme-based technology was applied in the FAM–ssDNA/pGO25000 system. Exo I is a sequence-independent 3′–5′ exonuclease that cleaves ssDNA. It has been reported that the digestion of Exo I is limited by binding to the targets to form the DNA duplex and G-quadruplex structures [11,14,15,32]. In this study, pGO25000 was synthesized as an efficient nanoquencher of fluorescence for the proposed strategy to detect Hg^2+^. After adding Exo I, the special enzyme efficiently digested the FAM–ssDNA in the direction of 3′ to 5′, and the fluorescence was restored. However, if dsDNA was present because of the formation of the T–Hg^2+^–T construct after adding Hg^2+^, which could suppress the activity of Exo I, no fluorescence was restored. Therefore, the fluorescence “turn-on” indicated that no Hg^2+^ was present in the analytical sample, and vice versa. It is expected that this strategy could provide a novel method to detect Hg^2+^ with great sensitivity and high selectivity.

### 2.2. Characterization of pGO25000

GO has good water solubility due to the abundance of hydrophilic groups (hydroxyl, carboxylic, epoxy) that have been introduced onto the surface of GO after a series of chemical modification processes. Because pGO25000 was prepared by grafting PEI onto GO nanosheets, pGO25000 also has good dissolvability. To study the surface charge of GO and pGO25000, zeta potential analysis was performed at concentrations of 1 mg/mL GO and pGO25000 solution. The GO solution showed a negative zeta potential level of −37.6 Mv, as shown in Figure 1, while pGO25000 had a positive zeta potential level of 25 mV because the PEI linkers completely changed the pGO25000 surface charge; thus, positively charged GO (pGO25000) was obtained [33]. To investigate the structural change in the condensation reaction of pGO25000 synthesis, Raman spectroscopy, FT-IR analysis, and high-resolution XPS were performed. Figure 2A shows the Raman spectra of GO and pGO25000, and two bands located at approximately 1320 cm^−1^ and 1596 cm^−1^ can be attributed to the D and G bands of graphitic materials, respectively. It is well known that the defect level of graphene sheets can be evaluated by the peak intensity ratio of the D band to the G band (I_D_/I_G_), and a higher I_D_/I_G_ commonly indicates an increase in the degree of disorder [34]. pGO25000 gave a higher I_D_/I_G_ ratio of 2.30 compared with GO (1.85), which can be attributed to the condensation reaction by incorporation of PEI, reducing the oxygen functional group and increasing the sp^3^ carbon form [35,36,37]. Figure 2B shows the FT-IR spectra of GO and pGO25000. Peaks located at ~1720 cm^−1^, ~1620 cm^−1^, ~1400 cm^−1^, and ~1090 cm^−1^ can be assigned to the stretching vibrations of the C=O, C–C, C–OH, and C–O (epoxy) groups [35,38,39]. Of note, the FT-IR spectrum of pGO25000 showed that PEI was successfully grafted onto the GO surface. Compared to GO, the N–C=O peak at 1650 cm^−1^ appeared with the disappearance of the C=O peak at 1720 cm^−1^ in pGO25000. Meanwhile, the C–O (epoxy) peak was replaced by the C–N peak (1384 cm^−1^) on pGO25000. The N–C=O and C–N groups were produced by the amine reacting with the COOH and C–O (epoxy) groups. The band at 1580 cm^−1^ appeared first, which corresponded to the C=N stretch by Schiff’s base reaction [40,41].

As shown in Figure 3A, there was almost no N1s signal in the spectrum of GO, whereas the spectrum of pGO25000 presented a clear N1s peak. After calibration of the binding energy position with C1s (284.4 eV) in XPS spectra, the five main peaks of carbon bonding in the C1s XPS spectra of GO with binding energies at 283.7, 284.4, 286.1, 286.9, and 288.3 eV (Appendix A) were attributed to the C=C, C–C, C–O (hydroxyl and epoxy), C=O, and C(O)O bonds, respectively [25,42]. After reacting with PEI, the signal at 285.3 eV (C–N bond) appeared along with the disappearance of the C(O)O bond signal, which indicated that the condensation reaction between the amino group and carboxyl group was completed. The peak at 286.0 eV (C–O) was dramatically decreased due to the epoxy reacting with PEI (Figure 3B). The N1s spectrum had fitted curves at 400.4, 399.1, and 398.2 eV (Figure 3C), corresponding to the binding energies of nitrogen atoms in NH_3_^+^, CONH, and PEI [40,43,44]. Compared to the O1s spectrum of GO (Appendix A), pGO25000 was deconvoluted into four peaks (Figure 3D), three of which were similar to those of GO, i.e., C=O (530.5 eV), C–OH (531.7 eV), and C–O (532.4 eV) [43]; a new peak with a bonding energy of 530.9 eV appeared, corresponding to the CONH bond. These results indicated that pGO25000 was successfully obtained by GO reacting with PEI.

### 2.3. Fluorescence Quenching of FAM–ssDNA by pGO25000

To better understand the fluorescence quenching efficiency of positively charged GO depending on the pH values, a solution with a pH range of 7.5–9.0 was investigated. The fluorescence intensity of FAM–ssDNA in a 600 nM Hg^2+^ solution significantly increased and then slightly decreased with increasing pH values, as shown in Appendix A. At pH 8.5, the maximum fluorescence signal was obtained. After adding 0.1 μg/mL pGO25000, there was a sharp reduction in fluorescence intensity due to fluorescence quenching caused by pGO25000, and there was no large difference under various pH values from 7.5 to 9.0. Hence, pH 8.5 was the optimum pH according to the ratio of the fluorescence intensity without pGO25000 and in the presence of pGO25000. It has been mentioned that fluorescence quenching by pGO25000 can be completed immediately; thus, fluorescence detection was instantly performed after adding pGO25000.

To demonstrate that the positively charged GO (pGO25000) is more efficient in fluorescence quenching for DNA probes, GO and pGO25000 were analyzed. The fluorescence of FAM–ssDNA was quenched by various concentrations of GO from 0.01 to 30 μg/mL, as shown in Appendix A. As the amount of GO increased, the fluorescent signal was reduced, and fluorescence quenching was not efficient even if the concentration was enhanced to 30 μg/mL. However, the fluorescence was almost quenched by pGO25000 at 0.1 μg/mL (Figure 4). Thus, positively charged PEI plays a crucial role in the affinity between pGO25000 and FAM–ssDNA, which can promote the fluorescent dyes to be close to pGO25000, thus increasing the FRET efficiency to quench fluorescence. The fluorescence spectra of FAM–ssDNA quenched by pGO25000 in the range from 0 to 30 μg/mL with excitation at 495 nm are shown in Figure 4 and Appendix A. The signal intensity of the FAM–ssDNA probes was moderately reduced with increasing pGO25000 concentration, clearly increased with a pGO25000 concentration higher than 0.1 μg/mL, and then decreased again until the concentration of pGO25000 was greater than 6 μg/mL. The fluorescence quenching ability of pGO25000 was induced by electrostatic and π–π stacking interactions, which were dependent on the concentration of pGO25000. This result indicated that pGO25000 had a better binding affinity with the FAM–ssDNA probes at very low concentrations from 0 to 0.1 μg/mL due to electrostatic interactions. As pGO25000 increased from 0.1 μg/mL to 6 μg/mL, fluorescence quenching was not efficient. This was possibly caused by the steric hindrance of PEI in pGO25000, which impeded the interaction of FAM–ssDNA with GO in pGO25000, reducing π–π stacking and hydrogen bonding between nucleobases and GO. However, the fluorescence quenching efficiency was improved when the pGO25000 concentration was more than 6 μg/mL. This phenomenon was observed because the electrostatic interaction between FAM–ssDNA and pGO25000 was increased, and fluorescence quenching was mainly dependent on the electrostatic interaction.

The influence of various metal ions on the fluorescence quenching ability of pGO25000 was also assessed by measuring the fluorescence intensity, and different metal ions were used, including Hg^2+^, K^+^, Sn^2+^, Al^3+^, Ni^2+^, Mn^2+^, Mg^2+^, Cu^2+^, and Co^2+^. Appendix A shows that there was no influence on the fluorescence quenching ability. Therefore, pGO25000 was synthesized as an efficient nanoquencher of fluorescence.

### 2.4. Fluorescence Detection of Hg^2+^

Fluorescence quenching efficiency is dependent on the interaction between DNA and GO, which is determined by the length of DNA [45,46], GO surface modification [25], size of GO [30], and concentrations of DNA and GO [47]. Compared to GO, the positively charged modified GO (pGO25000) had a perfect fluorescence quenching ability at very low concentrations. However, there was no large change in the fluorescence quenching of FAM–ssDNA by 0.1 μg/mL pGO25000 with or without Hg^2+^. However, FAM–ssDNA exists in the hairpin structure due to the formation of T–Hg^2+^–T after adding Hg^2+^. It was indicated that fluorescence quenching was efficient for the same sequences of DNA with different structures. Based on this result, an Exo I-assisted strategy to detect Hg^2+^ is proposed.

To demonstrate the effect of pH values on T–Hg^2+^–T complex formation and the activity of Exo I, the Exo I/pGO25000-assisted FAM–ssDNA sensor was studied for Hg^2+^ detection, as demonstrated in Appendix A. The fluorescent signal was significantly improved with increasing pH values in the absence of Hg^2+^. It is well known that FAM is a pH-dependent dye, and the optimal pH value is >8.5 [48,49]. FAM–ssDNA can be digested by the 3′–5′ Exo I and releases FAM dyes in a range of pH 7.5~9.0; thus, the fluorescence response can be recovered without Hg^2+^. When Hg^2+^ was present, all samples at various pH values were fluorescence-quenched except for the conditions at pH 7.5 and 9.0. Thus, the FAM–ssDNA probe could be subjected to the conditions at pH 8.0 and 8.5. The fluorescence signal was not restored because of the FAM–hairpin DNA structure formation, which was caused by the T–Hg^2+^–T construction, which prevented digestion by Exo I. However, at pH 7.5 and 9.0, the fluorescence was slightly recovered, and the reason could be attributed to the overactivity of Exo I under these conditions. As a result, 10 mM Tris–HNO_3_ buffer (40 mM NaNO_3_) with a pH value of 8.5 was used during Hg^2+^ detection.

### 2.5. Sensitivity of Hg^2+^ Detection

The Hg^2+^ concentration has a large effect on the FAM–ssDNA probe with the pGO25000/Exo I-assisted strategy. The sensitivity of this proposed method was determined using various concentrations of Hg^2+^ solution. Figure 5A shows that the emission signal was gradually reduced by excitation at 495 nm with increasing Hg^2+^ concentrations from 0 to 800 nM, and fluorescence was nearly quenched when the concentration of Hg^2+^ was greater than 600 nM. In a certain range, the higher the concentration of Hg^2+^, the more efficient the fluorescence quenching. This occurred because Exo I activity was restricted by the DNA hairpin structure, which was formed by the T–Hg^2+^–T complexes, and the FAM dye could not be released from the DNA strand, which was quenched by pGO25000. Without Hg^2+^ added, the fluorescence signal was perfectly recovered after digestion by Exo I. It was attributed to the destruction of the interaction between the DNA strand and pGO25000 during the hydrolysis of DNA; thus, the FAM dye was released from the DNA strand and kept far away from pGO25000; then, the fluorescence was restored.

The relative fluorescence intensity (F/F_0_) decreased proportionally as the Hg^2+^ concentration increased. It showed excellent analytical performance with a linear relationship in the range of 0 to 250 nM Hg^2+^, following a linear correlation equation described as y = −0.0031x + 0.9804 (y represents F/F_0_, x represents the concentration of Hg^2+^ in solution, R^2^ = 0.9955). Based on the 3σ slope, the limit of detection (LOD) for the Exo I/pGO25000-assisted FAM–ssDNA sensor was estimated to be 3.93 nM, which was far below the largest permissible dose of Hg^2+^ in potable water (10 nM) by the U.S. Environmental Protection Agency (EPA) [21,50]. The obtained result indicates that this biosensor strategy has potential applications in the quantitative analysis of Hg^2+^ at certain concentrations. Various nanomaterials were used as the nanoquencher of DNA probes, which were designed for Hg(II) detection, and the results are shown in Table 1. The sensitivity and fluorescence quenching efficiency of the nanoquenchers were compared using the previously presented analytical methods. Note that the proposed scheme presented a lower LOD and higher sensitivity compared with that using GO, SWCNTs, or MWCNTs as the nanoquencher, and pGO25000 had a more efficient fluorescence quenching ability compared to C_60_.

### 2.6. Selectivity of Hg^2+^

Confirming the selectivity for Hg^2+^ is the key point to evaluate the performance of the developed DNA biosensor. To estimate the selectivity of Exo-I/pGO25000 by the FAM–ssDNA sensing system, experiments were performed to detect the fluorescence intensities of the Exo I/pGO25000-assisted FAM–ssDNA probes in the presence or absence of Hg^2+^ (600 nM) solution mixed with other metal ions (K^+^, Fe^2+^, Sn^2+^, Al^3+^, Ni^2+^, Mn^2+^, Mg^2+^, Cu^2+^, and Co^2+^) at a concentration of 6 mM, as shown in Figure 6. Hg^2+^ produced a remarkable decrease in fluorescence intensity, indicating that only Hg^2+^ could bind to two thymine bases, and T–Hg^2+^–T mismatched base pairs formed, resulting in the formation of a hairpin structure; thus, degradation by Exo I was hindered. As a result, the FAM–hairpin DNA remained for fluorescence quenching by pGO25000. The obtained results also revealed that the proposed method could still detect Hg^2+^, which mixed with other probable interference metals. Clearly, the biosensor based on FAM–ssDNA probes and Exo I/pGO25000 has an excellent selectivity towards Hg^2+^.

### 2.7. Application in Real Samples

To assess the approach feasibility, the Exo I/pGO25000-assisted FAM–ssDNA biosensor strategy was used for the Hg^2+^ analysis of pond water samples collected from a pond at Zhejiang Sci-Tech University. The impurities of pond water were removed by filtration using NY 0.22 μm, and three concentrations of Hg^2+^ (25 nM, 50 nM, 200 nM) were diluted by the pond water, which were individually detected by the sensor system. The recoveries of Hg^2+^ in pond water were in the range of 99.6%–101.1%, and all relative standard deviations (RSDs) were as low as 5% (*n* = 3) (Table 2). It is revealed that the quantitative detection of Hg^2+^ can be performed by this sensitive biosensor method. Meanwhile, it illustrated that the proposed sensor has good feasibility and accuracy to measure the Hg^2+^ concentration in real samples.

## 3. Experimental

### 3.1. Chemicals and Materials

The oligonucleotide (FAM–ssDNA probe: 5′-FAM-TATCGTGCTCCCCTGCTCGTTA) was purified by HPLC and purchased from Sangon Biotech Co., Ltd. (Shanghai, China). The oligonucleotide stock solution (50 μM) was prepared with double distilled water (ddH_2_O). Exonuclease I (Exo I, 20 U/μL) was provided by Bio Basic Inc. (Canada). A 10 mM Tris–HNO_3_ buffer including 40 mM NaNO_3_ (pH 8.5) was used in the experiments. Hg^2+^ solutions with different concentrations were prepared from a standard mercury ion solution (5.0 mM), which was purchased from Aoke Biology Research Co., Ltd. (Beijing, China). All metal salts and tris(hydroxymethyl)aminomethane (Tris) were obtained from Macklin (Shanghai, China) and Aladdin (Beijing, China). Graphene oxide (GO) was supplied by XFNano Material Tech Co., Ltd. (Nanjing, China). PEI (average MW = 25,000) was purchased from Sigma-Aldrich. All other chemicals and reagents used in this work were of analytical grade and used without further purification. The pond water was prefiltered with an NY 0.22 μm filter.

### 3.2. Apparatus

Fluorescence spectra were measured by an F-4600 fluorescence spectrometer (Hitachi, Japan) at RT (room temperature, approximately 27 °C). The emission spectra data were acquired from 511 nm to 620 nm after illuminating at the maximum excitation wavelength of 495 nm, and the fluorescence intensity was measured at 520 nm (Em_max_). Both the excitation and emission slit widths were set at 5 nm and 10 nm, respectively. High-resolution X-ray photoelectron spectroscopy (XPS) was performed using a Thermo Scientific K-Alpha XPS spectrometer (Kratos Analytical, Manchester, UK). The charge polarity and density of GO/pGO25000 colloids were obtained by zeta potential measurement (Malvern Zetasizer Nano ZS90, Great Malvern, UK). The Raman spectrum was recorded by an Optosky ATP3007 (Xiamen, China) Raman spectrometer with a 785 nm excitation lase. A Nicolet iS50 spectrometer (Thermo Fisher Scientific, Waltham, USA) using KBr pellets was used for Fourier-transform infrared (FT-IR) characterization.

### 3.3. Preparation of pGO25000

Positively charged GO (pGO25000) was synthesized by grafting PEI (MW = 25,000) onto GO nanosheets via a condensation reaction between amino groups and carboxyl groups. Briefly, 120 mg of GO, 1.0 g of PEI, and 300 mg of EDC were sequentially dissolved in 40 mL of ddH_2_O. Then, the pH value of this mixture was adjusted to pH 7.0 by adding a certain amount of diluted HCl (1.0 M) and stored at 4 °C for 24 h. Then, the mixture was dialyzed against ddH_2_O for two weeks to remove foreign ions. After diluting this mixture with ddH_2_O and ultrasonication, the pGO solution (1 mg/mL) was successfully prepared.

### 3.4. Fluorescence Quenching Assay

A 50 nM FAM–ssDNA probe was incubated in 200 μL of Tris–HNO_3_ buffer (10 mM, 40 mM NaNO_3_, pH 8.5) with different concentrations of pGO25000/GO (0, 0.0001, 0.001, 0.005, 0.01, 0.02, 0.05, 0.1, 1, 3, 6, 15, and 30 μL/mL). The mixtures were immediately measured by a fluorescence spectrometer, and the fluorescence emission was monitored in the wavelength range from 511 to 620 nm. Each sample solution was repeated and measured at least three times.

### 3.5. Study of Exo I Activity in Hg^2+^ Detection

A 0.01 nmole FAM–ssDNA probe (50 μM, 0.2 μL) and 1 μL of Hg^2+^ solution (600 nM) were added to a 0.5 mL centrifuge tube, followed by dilution to 10 μL with Tris–HNO_3_ buffer (10 mM, 40 mM NaNO_3_, pH 8.5). The FAM–ssDNA probe/Hg^2+^ solution was added to 2 U of Exo I and incubated for 30 min at 27 °C (RT) or 5 min at 40 °C in an oven. Then, pGO25000 (0.1 μL/mL) was mixed with the FAM–ssDNA probe/Hg^2+^/Exo I mixture and diluted with Tris–HNO_3_ buffer (10 mM, 40 mM NaNO_3_, pH 8.5) to a final volume of 200 μL. Exo I nuclease was denatured by heating at 80 °C for 15 min before the fluorescence test. To evaluate the sensitivity of Hg^2+^ detection, the final concentrations of Hg^2+^ were 0, 2, 5, 10, 30, 50, 120, 150, 200, 250, 300, 400, 600, and 800 nM. To evaluate the selectivity of Hg^2+^ detection, 6 μM K^+^, Na^+^, Cu^2+^, Mn^2+^, Ni^2+^, Pb^2+^, and Fe^3+^ and 600 nM Hg^2+^ were used. The fluorescence emission was analyzed from 511 to 620 nm for all samples with an excitation wavelength of 495 nm, and the maximum fluorescence intensity was measured at 520 nm. To assess the application of the DNA sensor in the real samples, a 50 nM FAM–ssDNA probe was incubated in 10 μL of Tris–HNO_3_ buffer (10 mM, 40 mM NaNO_3_, pH 8.5) with three different concentrations of Hg^2+^ solution (25 nM, 50 nM, 200 nM), which were prepared from the pond water. Each sample solution was analyzed three times.

## 4. Conclusions

In summary, positively charged GO (pGO25000) was synthesized by modification of the GO surface with PEI and used in fluorescence quenching of a DNA probe. Compared to GO, the fluorescence quenching efficiency of pGO25000 was dramatically improved due to the additional electrostatic interaction induced by PEI. Electrostatic attraction plays a vital role in the interaction between pGO25000 and DNA strands, which increases the affinity of pGO25000 to the DNA strands. As a result, when pGO25000 is at a very low concentration (0.1 μg/mL), it possesses a higher and more efficient fluorescence quenching ability compared to GO. In view of the perfect fluorescence quenching efficiency of pGO25000, the pGO25000 and FAM–ssDNA probes were designed for Hg^2+^ detection with the assistance of Exo I, and fluorescence was specifically prevented by adding Hg^2+^. The FAM–ssDNA probe was formed in a hairpin structure in the presence of Hg^2+^ due to the formation of T–Hg^2+^–T complexes, which imposed restrictions on the degradation by Exo I. However, the fluorescence was recovered without Hg^2+^ because of the hydrolysis of FAM–ssDNA caused by Exo I. Therefore, the limit of detection of 3.93 nM was obtained. Compared to other metal ions, the Exo I/pGO25000-assisted FAM–ssDNA sensor has a great selectivity for Hg^2+^. In addition, the designed strategy is applicable for Hg^2+^ detection in real samples with satisfactory recoveries. In consideration of the efficient fluorescence quenching property of pGO25000, as well as the absolute quantification ability by the Exo I-assisted FAM–ssDNA sensor, the biosensor mechanism can be applied in more toxic substances and in gene mutation analysis (e.g., SNP).

## Data Availability

Not applicable.

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
