# Peer review of "Function of Graphene Oxide as the “Nanoquencher” for Hg^2+^ Detection Using an Exonuclease I-Assisted Biosensor"

_ijms, 2022, doi:10.3390/ijms23116326_

Round 1

Reviewer 1 Report

The article presents interesting results that open the way to the creation of biosensors sensitive to Hg2+ impurities in water. The manuscript is well structured, the results are presented in an understandable way, accompanied by well-designed diagrams and graphs. The article may be published after a minor revision.

Authors should:

1. to check the correspondence of the first use of the abbreviation and its decoding. For example, the abbreviation FRET first occurs in line 72, and its decoding is given in line 96. Some abbreviations are not decoded at all.

2. to provide an explanation for the phrase "pGO25000 gave a lower ID/IG ratio of 1.85 than GO (2.30), which can be attributed to the condensation reaction by reducing oxygen atoms and lessening sp3 carbon defects" lines 136-137. What is meant by "sp3 carbon defects" and why is the well-known C-V defect band at hv = 283.7 eV not taken into account in XPS spectra?

3. Scheme1 shows that PEIs are located on only one side of the GO nanosheets. Actually, it is not. Authors should provide clarification on this matter.

Author Response

Reviewer 1

  1. to check the correspondence of the first use of the abbreviation and its decoding. For example, the abbreviation FRET first occurs in line 72, and its decoding is given in line 96. Some abbreviations are not decoded at all.

I have corrected all the errors of the abbreviation and its decoding in Page 1-4, 10, Line 2, 12-15, 18-24, 44-47, 72-85,102, 204.

  1. to provide an explanation for the phrase "pGO25000 gave a lower ID/IG ratio of 1.85 than GO (2.30), which can be attributed to the condensation reaction by reducing oxygen atoms and lessening sp3 carbon defects" lines 136-137. What is meant by "sp3 carbon defects" and why is the well-known C-V defect band at hv = 283.7 eV not taken into account in XPS spectra?

We made a mistake, and we analyzed GO and pGO25000 by Raman spectroscopy again, pGO25000 gave a higher ID/IG ratio of 2.30 than GO (1.85), which can be attributed to the condensation reaction by incorporation of PEI, reducing oxygen functional group and increasing sp3 carbon form. The change was in Page 5, Line 143-146, 158. The binding energy calibration in XPS spectra (284.4 eV) has been done. The five main peaks of carbon bonding in the C1s XPS spectra of GO with binding energies at 283.7, 284.4, 286.1, 286.9, 288.3 eV (Fig. S1A) were attributed to the C=C, C-C, C-O (hydroxyl and epoxy), C=O, and C(O)O bonds, respectively; and The five main peaks of carbon bonding in the C1s XPS spectra of pGO25000 with binding energies at 283.6, 284.4, 285.3, 286.0, 287.0 eV (Fig. 3B) were attributed to the C=C, C-C, C-N, C-O (hydroxyl) and C=O bonds, respectively. The change was in Page 8, Line 165-178. The Fig. 2A, Fig. 3B and Fig. S1A have been updated.

  1. Scheme1 shows that PEIs are located on only one side of the GO nanosheets. Actually, it is not. Authors should provide clarification on this matter.

PEI are located on both sides of the GO nanosheets, We have updated the Scheme1 in Page 4 Line 94.

Reviewer 2 Report

The manuscript submitted by Jieqiong Qiua and co-workers describes a new nanomaterial used as a nanoquencher into a biosensors used for Hg2+ detection. To increase the selectivity the FAM-ssDNA probes were used with the assistance of Exo I.

 The manuscript is well written, and the conclusion are sustained by the experimental data. However, there are some issues that need to be addressed before being accepted:

1.       Please avoid abbreviations in title and abstract. All abbreviations need to be explained at their first use.

2.       Indeed, the method seems very sensitive towards Hg2+ ions. However, I think that the authors should compare the results obtained by their method with the other biosensors reported in the literature insisting on the advantaged of their method.

3.       The details regarding the analysis of real samples should be added in the experimental part.

Author Response

Reviewer 2

  1. Please avoid abbreviations in title and abstract. All abbreviations need to be explained at their first use.

I have already removed abbreviations in title and abstract, and corrected all the errors of the abbreviation and its decoding in Page 1-4, 10, Line 2, 12-15, 18-24, 44-47, 72-85,102, 204.

  1. Indeed, the method seems very sensitive towards Hg2+ ions. However, I think that the authors should compare the results obtained by their method with the other biosensors reported in the literature insisting on the advantaged of their method.

The comparison of our method and the other biosensors have been listed in Table 1 and the description has been added in Page 10-11 Line 278-284, 293-294.

  1. The details regarding the analysis of real samples should be added in the experimental part.

The details have been added in the manuscript in Page 14 Line 386-389.